# Recent Preclinical and Clinical Progress in Liposomal Doxorubicin

**DOI:** 10.3390/pharmaceutics15030893

**Published:** 2023-03-09

**Authors:** Kenan Aloss, Peter Hamar

**Affiliations:** Institute of Translational Medicine, Semmelweis University, 1094 Budapest, Hungary; kenan.aloss@phd.semmelweis.hu

**Keywords:** doxorubicin, PEGylated liposome, targeted delivery, pH-sensitive liposome, thermosensitive liposome

## Abstract

Doxorubicin (DOX) is a potent anti-cancer agent that has garnered great interest in research due to its high efficacy despite dose-limiting toxicities. Several strategies have been exploited to enhance the efficacy and safety profile of DOX. Liposomes are the most established approach. Despite the improvement in safety properties of liposomal encapsulated DOX (in Doxil and Myocet), the efficacy is not superior to conventional DOX. Functionalized (targeted) liposomes present a more effective system to deliver DOX to the tumor. Moreover, encapsulation of DOX in pH-sensitive liposomes (PSLs) or thermo-sensitive liposomes (TSLs) combined with local heating has improved DOX accumulation in the tumor. Lyso-thermosensitive liposomal DOX (LTLD), MM-302, and C225-immunoliposomal(IL)-DOX have reached clinical trials. Further functionalized PEGylated liposomal DOX (PLD), TSLs, and PSLs have been developed and evaluated in preclinical models. Most of these formulations improved the anti-tumor activity compared to the currently available liposomal DOX. However, the fast clearance, the optimization of ligand density, stability, and release rate need more investigations. Therefore, we reviewed the latest approaches applied to deliver DOX more efficiently to the tumor, preserving the benefits obtained from FDA-approved liposomes.

## 1. Introduction

Doxorubicin (DOX) (brand name Adriamycin) is a member of the anthracycline anti-cancer drug group and one of the most effective agents in this family. It was discovered in 1969 as a metabolite of a mutated Streptomyces peucetius [1]. DOX alone, or in combination with other chemotherapeutic drugs, is frequently used for the treatment of different cancer types, including breast [2], lung [3], ovarian [4], and bladder [5]. It exerts its action on cancer cells by intercalation into the DNA and the subsequent inhibition of topoisomerase-II-mediated DNA repair [6]. Moreover, free radical generation seems to play a significant role in DOX’s cytotoxicity [7].

Despite its efficacy, the use of DOX has been limited due to its adverse effects, such as cardiotoxicity [8] and myelosuppression [9]. Cardiotoxicity is the main dose-limiting side effect of DOX associated with its cumulative dose [8]. In the clinic, the cumulative DOX dose is limited to 400–450 mg/m^2^ [10]. However, subclinical cardiomyopathy can occur at lower cumulative doses (≈300 mg/m^2^) [11]. Cardiotoxicity severity may range from asymptomatic electrocardiography (ECG)-changes to decompensated cardiomyopathy with decreased left ventricular ejection fraction. There are three types of DOX cardiotoxicity: acute, early-onset chronic, and late-onset chronic [12]. Acute toxicity is less common, occurring during or within a few days after completion of treatment, and manifests as pericarditis, tachycardia, and electrocardiography changes, including repolarization problems such as T-wave flattening in the left chest lead (v3–6) and QT interval prolongation [13]. The early-onset chronic toxicity occurs within 1 year after DOX administration, while the late-onset chronic toxicity appears 1 or more years after treatment cessation [14]. Congestive heart failure is the main manifestation of chronic toxicity, with a 36% incidence risk at cumulative doses exceeding 600 mg/m^2^ [15]. 

The current drug delivery strategies of encapsulation into liposomes are promising tools to enhance DOX efficacy and improve the safety profile. Therefore, this review will focus on the strategies used to overcome the limitations of the approved liposomal DOX either by surface modification of these formulations or by triggering DOX release from the liposome in response to hyperthermia (HT) or pH. 

## 2. Limitations of the Approved Liposomal Doxorubicin Formulations

Two DOX-liposomal formulations have been authorized for clinical use: PEGylated liposomal DOX (PLD) (brand names: US: Doxil®, Lipodox®/generic Doxil/, Europe: Caelyx®) and non-PEGylated liposomal DOX (Myocet®) in Europe and Canada [16]. The clinically approved formulations reduced cardiotoxicity and hematological toxicity compared to free DOX [17,18,19]. PEGylation: the attachment of polyethylene glycol (PEG) has been applied to increase circulation half-life and reduce uptake by the reticulo-endothelial system (RES). PLD extravasates through the leaky vasculature of tumors resulting in enhanced delivery of PLD to the tumor. The main PLD toxicity is mucosal and cutaneous. The mucocutaneous toxicities are dose-limiting per injection; however, the reduced cardiotoxicity allows a larger cumulative dose [20]. The mucocutaneous toxicity is called hand-foot syndrome, also known as palmar-plantar erythrodysesthesia (PPE) [21]. The reason for this dose-limiting side effect is the PEG group. As PEGylation reduces cellular uptake [22,23], liposome accumulation increases in the capillaries of the skin [21,24]. In contrast, PPE syndrome was not observed in patients treated with the non-PEGylated liposomal DOX (Myocet), which confirms the role of PEG in this toxicity [25].

Even though the approved formulations provide a better safety profile, the therapeutic efficacy determined by progression-free survival (PFS) and overall survival (OS) is not superior to conventional DOX [17,18,19]. This non-superiority has been explained by several possible mechanisms: The poor release of DOX from the liposome [26]The entrapment of the liposome in the cellular lysosomes limits its delivery to the nucleus and reduces its bioavailable concentrations [27].These liposomes rely entirely on the enhanced permeability and retention (EPR) effect of the tumor. The liposomes should extravasate passively through the leaky tumor vasculature into the tumor interstitium (Figure 1A). This extravasation can be limited byThe notable heterogeneity of the EPR effect within the tumor and between tumor types [28]. Some tumor types, such as pancreatic cancer, exhibit poor EPR due to the thick fibrous tissue [29], and hematologic malignancies, such as leukemia and lymphoma, do not exhibit the EPR effect [30].The elevated interstitial fluid pressure (IFP) in most solid tumors [31].Interaction with extracellular matrix (ECM) components can hinder the distribution of the liposome to distant parts of the tumor [32].The lymphatic drainage in the metastatic microenvironment is not impaired as in the primary tumor; hence, the liposome is not entrapped in metastatic lesions. This fact explains the unsatisfactory effects of PLD and non-PLD in metastatic cancer [33].

Thus, new strategies are needed to overcome the limitations of the approved liposomes and improve DOX delivery to the tumor. 

## 3. Modified-PEGylated Liposomal Doxorubicin:

The insufficient cellular uptake and the accumulation of PLD in the skin necessitates further improvements, such as the modification of the PLD surface by molecules such as antibodies, peptides, aptamers, carbohydrates, or cell penetration enhancers (CPEs) (Figure 2). These approaches are gaining more interest as a promising strategy to enhance the efficacy of liposomal delivery [35]. These molecules are usually conjugated to the terminal end of the PEG chain and serve (except CPEs) as ligands for overexpressed targets on the surface of the cancer cells, increasing the specific accumulation of PLD in the tumor [36,37].

### 3.1. Antibody-Conjugated PLD

Antibodies were the first group of molecules that had been used for targeted drug delivery due to their high specificity and affinity [39]. They have been used for the functionalization of different liposomes [40,41,42], including PLD. Two antibody-conjugated PLDs have already reached clinical trials: (MM-302) [43] and C225-ILs-DOX [44].

MM-302 is a HER2-targeted antibody-PLD conjugate developed by Merrimack (MM) Pharmaceuticals and has been approved for clinical investigation in metastatic HER2-positive breast cancer. In preclinical models, MM-302 exhibited higher anti-tumor activity in comparison to PLD [45]. In addition, a combination of MM-302 and trastuzumab was synergistic in the inhibition of HER2-overexpressing tumors since these agents target different epitopes of the HER2 receptor [46]. Domunt et al. demonstrated that MM-302 was effective not only on the primary tumor but—unlike approved PLDs—also on the metastatic lesions in tumor-bearing mice. This in vivo study demonstrated that MM-302 significantly reduced the metastatic burden in the lung compared to PLD and resulted in higher accumulation in the diffuse and metastatic lesions where the lymphatic vessel density was high. The retention of MM-302 in the metastatic lesion was achieved by the interaction of HER2 with the anti-HER2 antibody [33]. This conclusion is promising because approved PLD and non-PLD have demonstrated insufficient efficacy in metastatic disease [17,47].

MM-302 proceeded to phase I clinical trial (NCT01304797) in 2011, where it was evaluated as monotherapy or in combination with trastuzumab and cyclophosphamide in patients with advanced HER2-positive breast cancer. Despite the efficacy of the treatment not being the primary endpoint of the study, MM-302 ≥ 30 mg/m^2^ alone or in combination demonstrated a better overall response rate (ORR) (i.e., 13%) and median PFS (i.e., 7.4 months) compared to treatment with physician’s choice (PFS= 3.3 months) [43]. This study revealed that MM-302 was well-tolerated in a dose of 8–50 mg/m^2^. The main adverse effects included fatigue, nausea, and neutropenia. Moreover, patients treated with MM-302 had less PPE syndrome than those treated with PLD in another clinical trial [43]. In the phase II trial: HERMIONE (NCT02213744), the efficacy of MM-302 combined with trastuzumab was evaluated versus the physician’s choice of chemotherapy (i.e., Gemcitabine, Capecitabine, Vinorelbine) plus trastuzumab. Unfortunately, the study was terminated in 2017 after a futility analysis showed that there was no improvement in PFS over the control groups. No recent updates have been published by Merrimack Pharmaceuticals regarding new clinical studies or the cessation of MM-302 development.

C225-immunoliposomal(IL)-DOX is a PLD conjugated with the antigen-binding fragment (Fab’) of the monoclonal antibody Cetuximab targeting the epidermal growth factor receptor (EGFR) [48]. In preclinical studies, C225-ILs-DOX exhibited superiority to PLD in tumor growth inhibition in different mouse models, including the MDA-MB-231 Vb100, a highly multidrug-resistant xenograft model [49,50]. Moreover, C225-ILs-DOX pharmacokinetics were comparable to PLD in healthy adult rats [49]. Based on these promising findings, C225-ILs-DOX had advanced to a phase I clinical trial (NCT01702129) in EGFR-overexpressing solid tumors where it was well tolerated at a 50 mg/m^2^ dose with the absence of cardiotoxicity and PPE syndrome. Furthermore, neutropenia was the most observed side effect. In contrast to preclinical studies, this trial revealed that the conjugation of the antibody to the PLD reduced the half-life of C225-ILs-DOX by 50% compared to PLD (31 h vs. 55–70 h) [44]. Given that EGFR is expressed in more than half of triple-negative breast cancers (TNBC), a phase II clinical trial (NCT02833766) was initiated in 2016 to test the efficacy of C225-ILs-DOX in TNBC. This study was terminated in 2020 after reaching the primary endpoint: PFS; however, no results have been published yet. Another phase I clinical trial (NCT03603379) was launched in 2018 to assess the pharmacokinetics of C225-ILs-DOX in patients with EGFR-positive, relapsed glioblastoma. Twenty-four hours after the intravenous administration of C225-ILs-DOX, only negligible concentrations of DOX were detected in the cerebrospinal fluid (CSF), indicating the inability of C225-ILs-DOX to cross the blood–brain barrier (BBB). On the other hand, resected glioblastoma tissues obtained from three patients exhibited high levels of DOX. Moreover, the patients did not experience any grade four or five adverse effects; however, severe pneumonitis was reported in one patient for the first time as an adverse effect of C225-ILs-DOX. Unfortunately, the results of this study were limited due to the small number (*n* = 9) of treated patients and the absence of a control group (i.e., free DOX) to confirm that the delivery of DOX to the tumor was achieved by C225-ILs-DOX and not by the DOX deliberated from the vehicle [51].

Further antibody-conjugated PLDs are under preclinical investigations at present. For instance, an anti-CD147-conjugated PLD (anti-CD147 ILs-DOX) was developed in 2018 to target the CD147, an overexpressed glycoprotein on the surface of hepatocellular carcinoma (HCC) cells that potentiates tumor progression. Treatment of human hepatocellular carcinoma (Huh-7) cells with anti-CD147 ILs-DOX resulted in a significant reduction in CD133-positive cancer stem cells compared to unmodified PLD, demonstrating the possibility of anti-CD147 ILs-DOX to target liver cancer stem cells. In addition, the biodistribution study revealed a higher accumulation of DOX in the tumor, liver, and spleen 24 and 72 h and in the lung 72 h after treatment with anti-CD147 ILs-DOX. Moreover, anti-CD147 ILs-DOX exhibited superior anti-tumor activity to the unmodified PLD in nude mice bearing the Huh-7 xenograft model [52]. 

Despite the high specificity associated with the use of antibodies for targeted therapy, they still have some limitations, such as the high production costs, the sophisticated preparation, immunogenicity, and the insufficient deep penetration due to their large size and high affinity that results in tight interaction with the target in the periphery of the tumor. Hence, other molecules have been considered for the decoration of the PLD surface to overcome those drawbacks [53].

### 3.2. Peptide-Conjugated PLD

Synthetic peptides as targeting ligands have some advantages over antibodies due to their small size, which enhances their penetration and distribution. Furthermore, unlike antibodies, peptides are non-immunogenic, which makes them a suitable approach for PLD modification [54,55]. Several peptides have been investigated to modify the PLD. Darban et al. modified the surface of PLD with leptin-derived peptides (i.e., LP16 and LP31) to target the leptin receptor, an overexpressed receptor in various cancer types. LP-PLD exhibited enhanced DOX cellular uptake in vitro associated with higher DOX tumor accumulation and anti-tumor activity in mice bearing C26 colon carcinoma compared to PLD [56,57]. The toxicity study revealed higher concentrations of LP-PLD (200 ligands) in the lung but not in the heart [56]. Similarly, Zahmatkeshan et al. functionalized the PLD with Anti-HER2/Neu peptidomimetic (AHNP), a specific ligand for HER2/Neu receptor. The functionalized PLD increased tumor growth inhibition and survival rate in mice bearing HER-2/neu-positive cells. Moreover, the survival rate correlated with the density of AHNP on the surface of PLD as the highest survival rate was observed after treatment with 100 AHNP-PLD and 200 AHNP-PLD [58]. 

An improvement in PLD efficacy can be achieved by PLD conjugation with peptides targeting cellular organelles such as the mitochondria, which regulate the cell death complex. Such a targeted PLD was developed in 2021 by modifying the PLD surface with Szeto-Schiller-02 (SS-02) peptide, a mitochondria-targeted peptide. The caspase activity assay demonstrated higher caspase-3 and caspase-9 activity in TUBO cells (a cell line highly overexpressing ratHER2/Neu), following treatment with SS-02-PLD bearing 200 ligands of SS-02 as compared to non-modified PLD. This in vitro finding was associated with stronger tumor growth inhibition and survival rate in mice bearing TUBO tumors. Moreover, there was a tendency for more DOX accumulation in the tumor, but the difference was not significant [59].

Recently, several peptide-conjugated PLDs have been developed. Avanzo et al. functionalized the PLD surface with LinTT1 peptide, a specific ligand for p32 protein overexpressed by cancer cells and cancer-associated cells such as M2-macrophages. Liposomes co-loaded with DOX and sorafenib (a multikinase inhibitor (MKI)) were used for breast cancer treatment. The LinTT1-conjugated liposome had a strong interaction with M2 primary human macrophages. As M2 macrophages accumulate in the hypoxic areas of the tumor, the strong binding of the LinTT1-conjugated PLD to M2 macrophages can lead to the accumulation of the liposome in the hypoxic central core of the tumor. Moreover, the LinTT1-conjugated liposome exhibited higher anti-cancer activity in 3D breast cancer spheroids compared to unconjugated liposomal delivery. Importantly, the prepared liposome was hemocompatible with less than 2% hemolysis. This hemocompatibility makes LinTT1-conjugated PLD safe for intravenous administration [60]. In another study, PLD was conjugated with a short peptide, namely P700, which has the ability to bind multiple tumor-overexpressed tyrosine kinase receptors, including vascular endothelial growth factor receptors (VEGFR 2, 3), platelet-derived growth factor (PDGFRa) and fibroblast growth factor receptor1 (FGFR1). The PLD-P700 resulted in great enhancement in cellular uptake and cytotoxicity in different cell lines, including mouse (4T1) and human (MCF-7) breast cancer cells and mouse endothelial cell line (H5V) compared to non-modified PLD [36]. A novel modified PLD was synthesized by Kato et al., who utilized cyclic RGDfK (cRGD), a cyclic peptide ligand that interacts with integrin αvβ3 on cancer cells. The cell viability assay on colon-26 cells exhibited much lower viable cells after treatment with RGDfK-PLD as compared to PLD [61]. However, the in vivo validation of these in vitro studies [36,60,61] has not been reported yet.

### 3.3. Aptamer-Conjugated PLD

Aptamers are short sequences of single-stranded DNA or RNA that represent a promising tool for active targeting and overcoming the difficulties with antibody-based systems due to the lack of immunogenicity, the simplicity of preparation and handling, and the small size that allows better penetration to tissues [62]. The aptamer is usually conjugated to the PEG chain, followed by post-insertion to the liposome. For example, Moosavian et al. linked PLD with a DNA aptamer, namely 5TR1-aptamer, that binds to Mucin1 (MUC1) receptor, an overexpressed receptor in various human adenocarcinomas. The biodistribution study demonstrated that the highest tumor level of DOX was achieved 48 h after 5TR1-PLD injection. The high tumoral DOX was associated with superior anti-tumor activity and survival rate in mice-bearing C26 colon carcinoma as compared to PLD [63]. Similar results were obtained by Mashreghi et al., who functionalized the surface of PLD with Syl3c aptamer to target the epithelial cell adhesion molecule (EpCAM), which correlates with poor prognosis when it is overexpressed. The conjugation was performed by EDC/NHS coupling chemistry to bind the amine group (-NH_2_) of the aptamer and carboxyl group (-COOH) of 1,2-distearoyl-sn-glycerol-3-phosphatidylethanolamine-*N*-PEG (DSPE-mPEG2000). The insertion of the aptamer into the liposome did not affect the stability of the PLD membrane in vitro. Despite the improvement in DOX accumulation in the tumor and survival rate, the tumor growth inhibition was not significantly enhanced in mice bearing C26 colon carcinoma compared to PLD [64].

RNA aptamers can also be used for the functionalization of PLD. Moosavian et al. linked PLD with the RNA aptamer: TSA14, which targets the HER2 receptor. The aptamer-conjugated PLD was more effective on tumor growth inhibition than non-modified PLD; however, no significant difference was observed in median survival time in mice bearing TUBO breast cancer [35]. Mashreghi et al. prepared an anti-EpCAM RNA aptamer conjugated PLD (ER-lip). A DSPE-mPEG2000-tethering DNA sequence was post-inserted to the surface of the liposome and linked with RNA aptamer. The modified PLD accumulated more in the tumor than PLD. Correspondingly, mice bearing colon carcinoma tumor model had the smallest tumor and the highest survival rate after treatment with the functionalized PLD [65].

### 3.4. Carbohydrates-Conjugated PLD

Carbohydrates are another class of molecules that have been used in the decoration of liposomes. Mannose is one of the most used carbohydrates for this purpose due to the high expression of mannose receptors on the immune cells, such as macrophages and dendritic cells. As these immune cells are abundant elements in the tumor microenvironment, targeting them can interact with immune response and tumor progression [66]. Cong et al. prepared PLD with the attachment of two ligands (i.e., D-mannose and l-fucose) to the PEG group. The aim of the dual targeting was first to increase PLD accumulation in the tumor by the interaction between l-fucose and e-selectin that is expressed on vascular endothelial cells (VECs) and second to enhance the PLD uptake by M2-like tumor-associated macrophages (M2-TAMs). The latter was based on the interaction of D-mannose with the mannose receptor, an overexpressed receptor on M2-TAMs. This dual modification increased cellular uptake in vitro. Moreover, the dual anchored PLD (DOX-MFPL) exhibited more tumor growth suppression in mice bearing S180 sarcoma attributed to the higher accumulation of DOX in the tumor and TAMs depletion. Interestingly, the safety profile of PLD was preserved after modification with no signs of cardiotoxicity, as demonstrated by the histological analysis of the heart [67].

### 3.5. CPE-Conjugated PLD

CPEs comprise a group of agents that can be linked to nanoparticles and promote their passage through the cell membrane and other cellular barriers in an ATP-dependent and independent manner [68,69]. Particularly, CPEs have positively charged amino acids, which facilitate their crossing of the negatively charged cell membrane [70]. CPEs can be of peptidic or non-peptidic nature. In this review, we will refer to both types as CPEs. The HIV-1 transactivator of transcription (TAT)-derived peptide is one of the most studied CPEs to modify the surface of PLD. Teymouri et al. linked the TAT peptide to DSPE-PEG2000 and was post-inserted into the liposome. Despite the enhanced cellular uptake in vitro, Jaafari’s group could not translate their results in vivo as the unmodified PLD was more effective on tumor growth inhibition in mice bearing C26 colon carcinoma tumors [71]. Zarazvand et al. explained this finding by the fast recognition and clearance of the TAT-PLD by the mononuclear phagocytic system (MPS) due to the attachment of TAT to the DSPE-PEG2000. Hence, TAT was attached to DSPE-PEG1000. As DSPE-PEG1000 is shorter than DSPE-PEG2000, it might reduce the exposure of TAT peptides in the blood. When compared to PLD, PLD functionalized with 100 and 400 ligands of TAT was more effective on tumor growth suppression in mice bearing C26 and B16F0 tumor models, respectively [72]. 

Non-peptidic CPEs have also been used for the decoration of the PLD. It was demonstrated in 2021 by Popilski et al. that the incorporation of tetraArginine-[G-1]-distearoyl glycerol (DAG-Arg4) into PLD surface resulted in higher DOX penetration into the cell nuclei in vitro. Although the DAG-Arg4-PLD did not enhance DOX accumulation in the tumor, it inhibited tumor growth significantly in mice bearing 4T1 tumors as compared to non-modified PLD. On the other hand, the main concern with CPE-conjugated PLD is that the CPE can enhance biodistribution to other organs and lead to systematic toxicity. Nevertheless, Stepensky’s group demonstrated that despite the high accumulation of DAG-Arg4-PLD in the lung, no enhanced distribution was observed in the heart [70]. 

Since several CPEs are nonselective to cancer cells, another strategy has emerged to dual modify the PLD surface with a CPE and a specific ligand for an overexpressed target. Deshpande et al. dual functionalized the PLD with transferrin (Tf) and a CPE, namely, octaarginine (R8). The aim was to target ovarian carcinoma cells overexpressing Tf receptors and enhance intracellular DOX delivery by R8. In vitro, the dual-modified PLD (Dual DOX-L) exhibited high cellular internalization with intense nuclear DOX. Moreover, treatment with Dual DOX-L resulted in the highest DOX tumor accumulation and tumor growth inhibition with negligible distribution to the heart in mice bearing A2780 ovarian xenografts [73]. Similarly, Amin et al. developed a dual-functionalized PLD using cyclic RGD peptide and TAT to increase the targeted area of the tumor. The dually modified PLD exhibited high vascular and cellular association in the tumor due to the RGD and TAT, respectively. Despite the low accumulation in the tumor, the functionalized PLD (D31-PLD) demonstrated the highest tumor growth inhibition and survival rate in mice bearing B16 tumors. However, the existence of both peptides on the surface of the PLD provoked fast clearance by the RES. This clearance was proportional to the density of TAT (i.e., 100 or 200 ligands), reducing the possibility of using this formulation for DOX delivery in the clinic [74].

In addition to their ability to penetrate the cell membrane, some CPEs, such as Cys-Gly-Lys-Arg-Lys (CGKRK) peptides, are also targeting ligands. CGKRK can target the P32 protein on tumor parenchymal and endothelial cells disrupting the role of P32 in angiogenesis. Therefore, CGKRK has been used to modify the PLD and improve PLD efficacy. The modification of PLD with 100 ligands of CGKRK (PLD-100) enhanced the anti-angiogenic properties in vitro compared to PLD. This finding was translated to superior tumor growth suppression and median survival time in mice bearing 4T1 tumors. Moreover, the CGKRK-modified PLD was safe, with no histological damage in major organs [75].

Despite the promising preclinical results with the functionalized PLDs (summarized in Table 1), some challenges need to be addressed, such as the reliance on the EPR effect, the heterogeneous expression of the targets [74], and the interaction with serum proteins [76]. Additionally, the ligand density to achieve efficient cellular uptake with low aggregation need to be optimized [76]. Thus, the most promising approach is to trigger DOX release from the liposome in response to a trigger such as HT, pH, light, or redox state. In the second part of this review, we will focus on temperature-sensitive liposomes because HT is already approved for clinical application. In addition, we will summarize the recent formulations of pH-sensitive liposomes for DOX delivery.

## 4. Thermosensitive Liposomal Doxorubicin

### 4.1. Traditional Thermosensitive Liposome (TTSL)

Thermo-sensitive liposomes (TSLs) are spherical vesicles composed of phospholipids that undergo a phase transition from a solid gel to a liquid-crystalline phase at a temperature above physiological temperature called melting phase transition temperature (Tm) [34]. In the gel phase, the phospholipids are well-arranged and immobile with fully extended hydrocarbon tails, preserving the liposome membrane impermeability. When the temperature approaches the Tm, the phospholipid heads become mobile, and the trans to a gauche shift in the configuration form of the C-C single bonds in the hydrocarbon chains takes place [79]. In other words, at the gel phase, the rotation around carbon bonds is restricted, and the hydrocarbon chains are in trans conformation (i.e., the dihedral angle = 180°). The trans conformers allow the hydrocarbon chains to pack tightly together. When the temperature approaches the Tm, the carbon atoms in the hydrocarbon chains become less restricted to move and rotate 120° relative to trans conformation resulting in the formation of gauche conformers (Figure 3). As the number of gauche conformers increases in the lipid hydrocarbon chains, the lipid pack becomes loose [80]. At this point, leaky and highly disordered microscopic regions start to form at the interface between the membrane domains that have become liquid and the ones that are still in the gel phase. Those permeable regions are called grain boundaries (Figure 4A). At a temperature higher than the Tm, the liposome membrane becomes fully fluid and permeable resulting in drug release [79,81].

The main component in all TSL formulations is a 1,2-DiPalmitoyl-sn-glycero-3-PhosphoCholine (DPPC) (Tm = 41.4 °C) which is usually mixed with small amounts of phospholipids with a higher Tm such as 1,2-DiStearoyl-sn-glycero-3-PhosphoCholine (DSPC) (Tm = 54.9 °C) to increase the membrane stability. The difference between the lower Tm DPPC and the higher Tm DSPC is that in DPPC palmitoyl, whereas in DSPC stearoyl, the residue is included. This explains the higher Tm in DSPC due to the longer hydrocarbon chain in the stearoyl residue [84]. The DPPC + DSPC mixture was the first described TSL formulation, developed by Yatvin et al. in 1978 and known now as traditional thermosensitive liposomes (TTSL) [85]. The main drawback of this formulation was the low rate and amount of release [85]. Therefore, Yatvin’s work was followed by several attempts to increase the membrane permeability by the inclusion of other phospholipids. Gaber et al. tested a combination of DPPC, hydrogenated soy phosphocholine (HSPC), cholesterol, and PEG at a molar ratio of (100:50:30:6). This mixture enhanced DOX release slightly (≈60% within 30 min), but the thermal range of phase transition was 42–45 °C which is not achievable clinically by mild hyperthermia (39–42 °C) [86]. Therefore, the aim in the more advanced TSL strategies is to enhance DOX release by the modification of the lipid bilayer using lysolipids, synthetic polymers and phosphatidylglycerol lipids.

### 4.2. Lyso-Thermosensitive Liposome (LTSL)

Lyso is a prefix that refers to the removal of one of the two fatty acid chains in phospholipids by hydrolysis [87]. Thus, lysolipids are small bioactive lipid molecules that contain only one acyl chain instead of two, such as Mono-Stearoyl-2-hydroxy-sn-glycero-3-PhosphoCholine (MSPC) and Mono-Palmitoyl-2-hydroxy-sn-glycero-3-PhosphoCholine (MPPC) [88]. The difference with DPPC/DSPC making up TTSL is that 2-hydroxy-sn-glycero-3-PhosphoCholine part is complexed with 1-stearoyl and 1-palmitoyl residues instead of two in DPPC/ DSPC. Due to the shorter hydrocarbon chain in palmitoyl, the addition of a small amount of MPPC leads to a lower peak phase transition compared to MSPC (40.5 °C vs. 41.3 °C) [89]. 

Incorporation of these shorter lysolipids into the TSL membrane dramatically increases the DOX released. It is postulated that lysolipids accumulate in the grain boundaries and result in stable pores in the lipid bilayer during the phase transition, leading to high membrane permeability and rapid DOX release from the liposome at the heated tumor (i.e., ≈80% of DOX released at 40–42 °C) (Figure 4B) [88,90]. This idea was first introduced in 1999 by Needham, who added 10% of MPPC into PEGylated DPPC membranes of TTSL (DPPC:MPPC:DSPE-PEG-2000, molar ratio 90:10:4) [91,92]. After that, Needham et al. and Kong et al. evaluated this formulation in mice bearing human squamous cell carcinoma xenograft (FaDu) and demonstrated higher DOX tumor accumulation and tumor growth inhibition after treatment with LTSL at 42 °C compared to treatment with TTSL and non-TSL liposome (NTSL) [92,93].

In contrast to NTSL, LTSL releases DOX into the bloodstream in the heated tumor, followed by DOX diffusion from the blood vessels into the tumor interstitium (Figure 1B). This intravascular release approach increases DOX accumulation in the tumor and bypasses the dependence on the EPR effect [94]. A histological analysis conducted by Manzoor et al. demonstrated that administration of LTSL+ HT doubled DOX penetration compared to Doxil + HT [94]. Moreover, Li et al. introduced a novel two-steps hyperthermia to maximize the benefits of the TSLs by firstly enhancing the tumor vasculature permeability and, as a result, the accumulation of the TSL in the tumor and secondly activating DOX release from the extravasated TSL to achieve higher interstitial levels. However, the two-step HT combined with the extravasation-based TSL was less effective on tumor growth inhibition in mice bearing BLM melanoma as compared to a combination of a one-step HT and TSL that provided fast intravascular release [95]. This conclusion was confirmed recently by Al-jamal et al., who investigated the additive effect of 2-HT on the efficacy of LTSL, TTSL, and NTSL in the human breast MDA-MB-435 xenograft model. After a single HT, LTSL was the most effective in suppressing tumor growth, with the longest median survival. Interestingly, applying the second HT 24 h after the first one did not improve either DOX tumor levels or the therapeutic effect of any of the administrated liposomal formulations as compared to the single HT approach. The lack of improvement in tumoral DOX and tumor growth inhibition was attributed to the enhanced blood perfusion in the heated tumor after the second HT. This enhanced perfusion can lead to DOX washing out from the tumor [96]. 

The main drawback of the LTSL is the possible dissociation of the lysolipids from the liposome membrane due to their desorption into biological components such as serum proteins. Lysolipid dissociation can lead to undesirable leakage at body temperature, followed by systematic toxicity and alteration in the thermosensitivity. Banno et al. demonstrated that LTSL lost ≈70% of lysolipids within one hour after injection into the circulation at 37 °C [97]. Moreover, LTSL released ≈ 35% of the encapsulated cargo within 1 h of incubation in fetal bovine serum (FBS) at 37 °C [98]. Needham attributed this DOX leakage from LTSL to H+ ions leakage through the membrane grain boundary defects. H+ ions leakage disrupts the protonated–unprotonated DOX balance inside the liposome and results in leakage of unprotonated soluble DOX [99]. 

The inclusion of 5–10 mol% cholesterol into the LTSL membrane has been proposed as a possible solution to stabilize the LTSL and reduce premature leakage. Sadeghi et al. exhibited that LTSL containing 5 and 10 mol% cholesterol was more stable than conventional LTSL, with approximately 13% leakage after incubation in FBS at 37 °C. Importantly, the incorporation of low amounts of cholesterol kept the fast-release kinetics feature of LTSL with complete DOX release within 2 min incubation in FBS at 42  °C [98]. Thus, the cholesterol-containing LTSL can provide some advantages over the conventional LTSL; however, more in vivo studies are required to investigate the effect of cholesterol inclusion on circulation kinetics and tumor accumulation.

Due to the premature leakage and short circulation time, the timing of LTSL administration (i.e., before or during HT) is crucial to achieving the possible clinical benefits of LTSL. Most of the previous preclinical studies applied HT immediately after [96,100] or shortly before LTSL administration [101,102]. Ponce et al. demonstrated that tumoral DOX was double when LTSL was injected during HT versus 15 min before HT [101]. This improvement in tumoral DOX was associated with better anti-tumor activity in rats bearing fibrosarcomas [101]. In addition, LTLD infusion during HT achieved higher DOX accumulation in the bladder wall of pigs bearing bladder cancer compared to DOX + HT [103]. 

Thermodox (LTLD) is an LTSL that was developed by Celsion corporation using the same formulation of Needham [92] with slight modification (DPPC:MSPC:DSPE-PEG-2000, molar ratio 86:10:4). LTLD is the first TSL to enter human clinical trials where it was used in combination with Radiofrequency ablation (RFA), a monotherapy for the treatment of small tumors (˂3 cm) in Hepatocellular carcinoma (HCC). The phase I clinical trial using this combination was conducted on 24 patients with HCC. This study exhibited a dose-response relationship of LTLD regarding the time until the failure of the treatment. Interestingly, most of the patients had tumors ˃ 3 cm. Following the promising results of the phase I study, FDA permitted Celsion to progress directly to the phase III study called HEAT [104]. 

The aim of the HEAT study (NCT00617981) was to evaluate the additive effect of a single dose (50 mg/m^2^) of LTLD administrated 15 min prior to RFA as compared to RFA alone in HCC patients with tumors 3–7 cm. Unfortunately, the study failed to reach its primary endpoint (i.e., PFS) as both PFS and OS rates were similar in both groups [105]. However, a subgroup analysis exhibited a significant enhancement in OS in patients who received LTLD + RFA ≥ 45 min in comparison to RFA alone. In contrast, OS was similar in both groups when RFA was applied ˂45 min [106]. Based on that, a second phase III study, the OPTIMA trial, was launched, where HCC patients were treated for a minimum of 45 min with RFA, and OS was chosen as the primary endpoint (NCT02112656). In 2021, Celsion corporation declared the cessation of the study because futility criteria were met at a planned interim analysis [107].

The similarities in the design of phase III trials might be responsible for the insufficient efficacy of LTLD. These similarities include using the same cancer type (i.e., HCC), the single administration of LTLD, and the application of RFA to trigger DOX release. Hence, the necessity for some modifications in future trials with LTLD is justified. These modifications can be (i) the utilization of a cancer type that is more sensitive to DOX (i.e., breast cancer), (ii) testing the multiple-dosage regimen of LTLD, (iii) the selection of other HT approaches to activate LTLD [108], such as magnetic resonance-guided high intensity focused ultrasound (MR-HIFU), an HT modality that can be used to perform tissues ablation (˃60 °C) [109] or mild HT (≈42 °C) [110], and modulated electro-hyperthermia (mEHT) that heats the tumor selectively (42 °C) [111,112]. Most of these aspects have been considered in the ongoing clinical trials with LTLD in different solid tumors (NCT04791228, NCT02536183, NCT03749850) (Table 2).

The ongoing i-Go trial (NCT03749850) is the first study to combine LTLD with MR-HIFU in breast cancer and the second study in breast cancer after the DIGNITY study (NCT00826085), which combined LTLD with microwave HT. The aim of the i-Go study is to replace DOX in the AC chemotherapy regimen (i.e., DOX+ cyclophosphamide) with a combination of LTLD and MR-HIFU (60 min, 40–42 °C). Thus, 12 patients with de novo stage IV her2-negative breast cancer will receive six cycles of LTLD combined with MR-HIFU and followed by cyclophosphamide. While the primary endpoints are the tolerability and feasibility in terms of the number of cycles and HT duration that can be completed, treatment efficacy represented by the radiological response is the secondary endpoint.

LTLD has the same toxicity profile as DOX in terms of hematological and gastrointestinal adverse effects. Alopecia, leukopenia, and neutropenia (grades 3 and 4) are the most frequently observed [106,113]. No dose-limiting cardiotoxicity was reported after LTLD treatment except few cases of asymptomatic declines in left ventricular ejection fraction (LVEF) after six cycles of LTLD [113]. However, most of the previous preclinical and clinical studies have administrated LTLD in a single dose. Thus, more evaluation of the cardiac function after repeated dosing of LTLD is required.

Similar to Doxil and Myocet, LTLD-induced hypersensitivity reaction was observed in dogs and pigs. Therefore, a prophylactic regimen consisting of corticosteroids (i.e., Dexamethasone) and antihistamines (H1 and H2) was applied before LTLD infusion [102,103,114]. This premedication, accompanied by LTLD slow infusion over 30 min, successfully eliminated the hypersensitivity in these animals and thus applied in the clinical trials [106,113]. On the other hand, the hypersensitivity reaction was avoided in many mice studies by using nude mice [92,100,115,116].

### 4.3. Polymer-Modified Thermosensitive Liposome (pTSL)

Another strategy to achieve thermosensitivity in the liposome membrane is the inclusion of thermosensitive polymers. Such polymers are characterized by their lower critical solution temperature (LCST), at which a coil−globule transition and phase separation occurs [117]. At temperatures below the LCST, the polymers are in a water-soluble coil state, preserving the liposome membrane stability and preventing drug release. As the temperature exceeds the LCST, the polymers start to lose their hydrogen bonds with water molecules and precipitate into a dehydrated globular form, resulting in the disruption of the liposome membrane and the release of its cargo (Figure 5) [118]. The desired LCST of the polymer (e.g., 38–42 °C) can be obtained by copolymerization with hydrophilic or hydrophobic co-monomers that can elevate or reduce the LCST, respectively [117,119].

While TTSLs release their cargo slowly and insufficiently and LTLD shows premature leakage at physiological temperature, the polymer-modified thermosensitive liposomes (pTSLs) could find the balance between TTSL and LTLD and overcome their limitations. Particularly, pTSLs display a more preferred balance between stability and DOX release profile. It has been found that at physiological temperature, pTSL was more stable than LTLD, releasing ˂7% of DOX within 90 min in serum [121]. Moreover, pTSL exhibited a rapid and higher release rate at 42 °C compared to TTSL (≈70% within 3 min) [122].

Different pTSL formulations have been investigated to encapsulate DOX. Poly (*N*-IsoPropyl-acryl-AMide) (pNIPAM) is the most used thermosensitive polymer for this purpose. PNIPAM is usually copolymerized with co-monomers to increase its LCST [123]. Terence et al. synthesized a DOX-pTSL by attachment of a copolymer composed of pNIPAM and pH-responsive PropylAcrylic Acid (PAA) to TTSL. The TTSL was composed of DPPC:HSPC:CHOL:DSPE-PEG-2000 at molar ratios: 100:50:30:6, respectively. The prepared pTSL exhibited a dramatic reduction in the thermal dose following 5 min incubation in HEPES buffer at 43 °C compared to TTSL. Additionally, the thermal dose was further decreased under slightly acidic conditions. This finding can be exploited clinically due to the acidic environment in the tumor [121]. Moreover, pTSL combined with FUS (43 °C, 5 min) was more effective on tumor growth inhibition in rats bearing tumors with greater penetration to the tumor center compared to free DOX and TTSL [122].

Mo et al. copolymerized pNIPAM with *N*-(2-HydroxyPropyl) MethacrylAmide (HPMA). The strong hydrophilicity of HPMA can increase the LCST of pNIPAM. Thus, the poly (NIPAM-r-HPMA) was incorporated into TTSL to prepare DOX-pTSL, which exhibited phase transition at 42 °C. The resulting pTSL had high stability at 37 °C while releasing around 70% DOX within 1 min at 42 °C. Moreover, the pTSL enhanced the cellular uptake in vitro and the tumor penetration in vivo as compared to TTSL. Interestingly, applying 5 min HT 24 h post-injection of pTSL was enough to exhibit superior anti-tumor activity in 4T1-bearing mice as compared to TTSL+HT with no cardiotoxicity. The shortly applied HT is highly important in the clinic as it results in less damage to normal tissues [120].

Another polymer that has been used for the modification of TSL is poly(2-(2-EthOxy) EthOxyethyl Vinyl Ether) (EOEOVE), which displays an LCST of around 40 °C. Poly (EOEOVE) has a greater ability to sensitize the liposomes to temperature than (pNIPAM) [124]. Kono et al. copolymerized poly(EOEOVE) with OctaDecyl Vinyl Ether (ODVE) that acts as a moiety anchor for stronger attachment of poly(EOEOVE) onto the liposome membrane. The resulting poly (EOEOVE)-OD4 copolymer was used to prepare pTSL for DOX. The poly(EOEOVE)-OD4-TSL was stable at 37 °C with less than 5% leakage. However, this pTSL exhibited significant DOX release at temperatures above 40 °C (50% at 43 °C, 85% within 1 min at 45 °C). In addition, the poly(EOEOVE)-OD4-TSL combined with HT (10 min, 45 °C) had a greater effect on tumor growth suppression in mice bearing C26 colon carcinoma than non-modified TSL [125].

Elk et al. encapsulated DOX into liposome grafted with copolymer composed of cholesterol and poly(*N*-(2-Hydroxypropyl) MethacrylAmide mono/dilactate) (chol-pHPMAlac). Chol-pHPMAlac exhibits tuneable critical solution temperature behavior. The ratio between pHPMmonolactate and pHPMdilactate was 43:57 [126]. The higher content of pHPM dilactate can reduce the LCST of the polymer due to the higher hydrophobicity of pHPM dilactate over pHPM monolactate. Moreover, cholesterol serves as an anchor to fix the polymer onto the liposome membrane [127]. The in vitro study revealed that (chol-pHPMAlac) TSL did not induce platelet activation in whole blood, which makes this TSL safe for intravenous administration. However, the complete release of DOX was obtained at 47 °C or higher, making (chol-pHPMAlac) TSL unsuitable for application with mild HT modalities [126].

The use of a variety of temperature-responsive copolymers in previous studies has resulted in pTSLs with different DOX release behavior and variance in the HT time required for the complete release. Therefore, these factors need to be optimized before reaching the clinical investigation.

### 4.4. Phosphatidylglycerol-Based Thermosensitive Liposome (DPPG-TSL)

DPPG-based TSLs are a new generation of long-circulating TSLs that combines both stability and rapid release. DPPG-TSL was first introduced by Lindner et al., using 30% of 1,2-dipalmitoyl-sn-glycero-3-phosphodiglycerol (DPPG2) in combination with DPPC and DSPC, omitting PEG and lysolipids [128]. Despite the absence of PEG, the long circulation of DPPG-based TSL was achieved by the inclusion of the synthetic lipid DPPG containing free hydroxyl groups, which presents strong hydrophilicity and inhibits interactions with serum proteins [128]. In contrast to lysolipids, DPPG does not affect the stability of TSL at physiological temperatures. It was shown that DOX leakage from DPPG-TSL did not exceed 5% after 1 h of incubation in serum at 37 °C compared to approximately 30% leakage from LTLD. Furthermore, DPPG-TSL exhibited a DOX release profile similar to LTLD at 42 °C in vitro, indicating that the fast and complete release of DOX can be achieved without the inclusion of lysolipids [129,130].

Due to the favorable pharmacokinetics, DPPG-TSL showed promising results in preclinical studies in different animal models. Hossann et al. revealed that DOX accumulation was higher in tumors but lower in the heart of sarcoma-bearing rats after treatment with DPPG-TSLDOX+ HT compared to LTLD+ HT. Consequently, rats treated with DPPG-TSL had the highest survival rate [130]. Similarly, DPPG-TSLDOX was evaluated in cats suffering from feline sarcoma [130,131]. Importantly, DPPG-TSLDOX administration was safe without serious toxicity [131]. In addition, a combination of DPPG-TSLDOX and HT exhibited a better response than conventional DOX, as demonstrated by the metabolic response determined with 18F-FDG-PET/MRI and histopathological analysis after tumor resection [130]. 

DPPG-TSLDOX was investigated as a treatment for muscle-invasive bladder cancer in rats and pigs [132,133]. Valenberg et al. evaluated DOX accumulation in the bladder wall of pigs treated with 20 mg/kg and 60 mg/kg of DPPG-TSLDOX and free DOX. The highest DOX accumulation in the bladder wall was observed after a combination of DPPG-TSLDOX and HT. On the other hand, lower DOX accumulation was detected in the heart and kidney of (DPPG-TSLDOX + HT)-treated pigs [132]. Consistent with this finding, rats treated with DPPG-TSLDOX + HT demonstrated higher complete tumor response than free DOX-treated rats (70% vs. 18%) [133].

Recently, a clinically feasible protocol has been developed to trigger DOX release from DPPG-TSL using MR-HIFU modality in healthy landrace pigs. Specifically, 50 mg DOX per m^2^ of DPPG2-TSL-DOX was infused for 30 min, followed by two local HT treatments initiated at 10 min and 60 min after the beginning of DPPG2-TSL-DOX infusion. The temperature of the heated muscle (i.e., thigh muscle) was kept at 42 °C for 30 min. Importantly, DOX quantification revealed much higher DOX in the heated muscle compared to unheated muscles. Moreover, low DOX concentrations were detected in the heart of the animals; however, cardiotoxicity was not evaluated in this study [134]. This study represents a step toward the clinical translation of DPPG-TSL-DOX. According to Regenold et al., DPPG-TSL-DOX is currently in clinical development by Thermosome GmbH in Germany [107].

### 4.5. Multifunctional-Thermosensitive Liposome

Applying one strategy to deliver DOX to the target tumor might not be enough. The difficulties with demonstrating significant clinical benefits from the LTLD in phase III clinical trials support this presumption. DOX can be delivered more efficiently to tumors by loading it in liposomes that combine the thermosensitivity with other strategies. Similar to the functionalized PLDs, TSLs can be surface modified with some molecules, such as targeting ligands. For instance, an LTSL modified with iRGD tumor-homing peptide was prepared and combined with HIFU for 10 min at 42 °C. Interestingly, iRGD-LTSL-DOX+ HIFU was more effective on tumor growth suppression in mice bearing the 4T1 model than LTSL-DOX+ HIFU. This finding was attributed to the ability of iRGD to selectively bind the αν integrins on the tumor angiogenic endothelial cells [135]. Similar results were obtained after DOX encapsulation in LTSL modified with a tumor-homing peptide, namely Cys-Arg-Glu-Lys-Ala (CREKA), a peptide that targets the clotted plasma proteins on the tumor vessel walls and stroma [116].

Lin et al. conjugated DOX with a CPE and encapsulated the conjugate along with magnetic fluid Fe_3_O_4_ in a TSL. The TSL was composed of DPPC: MSPC: DSPE-PEG2000 in a molar ratio of 87:3:10. The aim of the Fe_3_O_4_ was to generate heating (42 °C) under the application of an AC magnetic field. The magnetic TSL (DOX-CPE TSML) was able to release more than 80% of the cargo within 30 min at 42 °C with 7% leakage at 37 °C. Under the magnetic field, DOX-CPE TSML exhibited superior tumor growth inhibition to a non-magnetic DOX-CPE TSL activated by HT in MCF-7 tumor-bearing mice. Significantly, both formulations did not exhibit systematic toxicity demonstrated by the body weight [136]. In another study, Dorjsuren et al. synthesized a TSL consisting of DPPC, DSPE-PEG2000, and MPPC. The prepared TSL was loaded with DOX and Fe_3_O_4_ magnetic nanoparticles (MNP). After that, the TSL was conjugated with Cetuximab to target EGFR-expressing breast cancer cells. The resulting TSL revealed an increased DOX release under near-infrared (NIR) laser irradiation and acidic pH. Moreover, when exposed to NIR irradiation in vivo, the TSL elevated the temperature of the tumor surface to 48.7 °C due to the presence of the MNP. However, the in vivo anti-tumor activity of the prepared TSL has not been investigated [137].

A multifunctional pTSL was developed by Kono et al. by the incorporation of trastuzumab (HER) to target the HER2-positive cells and a fluorescence dye for NIR imaging to follow the biodistribution of the liposome. The HER-TSL accumulated more in the tumor in SK-OV3-bearing mice than the non-modified TSL. When combined with HT (44 °C) for 10 min, HER-TSL exhibited better anti-tumor activity than the non-modified TSL [138]. 

Alawak et al. designed a novel multifunctional magnetic DOX-TSL conjugated with MAB 1031 antibody to target the transmembrane metalloprotease-disintegrin (ADAM8) protein which exhibits high expression in triple-negative breast cancer (TNBC) cells and contributes negatively in tumor progression. In addition, gadolinium (Gd), a paramagnetic agent, was incorporated into the liposome to increase the contrast and cell detection by ultra-high field MR imaging (UHF-MRI). The incorporation of MAB antibody enhanced the in vitro cell-binding efficiency of MDA-MB-231 cells compared to non-modified TSL. In addition, MDA-MB-231 cells treated with MAB-TSL revealed higher viability reduction after 1 h of exposure to UHF-MRI. Moreover, the liposome was hemocompatible and safe for intravenous administration [139].

## 5. pH-Sensitive Liposome (PSL)

The tumor microenvironment (TME) has lower pH (pH: 6.5–6.8) compared to normal tissues (pH: 7.4). In particular, the pH of the endosomes and lysosomes is much more acidic (pH: 5-6). These pH differences can be exploited to develop acidic-responsive nanoparticles such as the pH-sensitive liposome (PSL) [140]. The pH sensitivity can be achieved by the inclusion of pH-sensitive lipids, pH-sensitive peptides, or pH-sensitive polymers. The most studied PSLs are the long-circulating PSLs which will be the focus of this section. Phospholipids derived from phosphatidylethanolamine (PE), such as dioleylphosphatidylethanolamine (DOPE), represent the main component of the long-circulating PSLs. DOPE is usually stabilized using carboxylated lipids, such as cholesteryl hemisuccinate (CHEMS) [141]. At physiological pH, CHEMS is ionized and packed between the DOPE molecules, stabilizing the liposome. Following the endocytosis of the PSLs, CHEMS molecules become protonated due to the low pH of endosomes. Consequently, the liposome is destabilized, releasing the encapsulated DOX [142]. Silva et al. exhibited that PSL-DOX had higher accumulation in the tumor than a non-PSL-DOX composed of HSPC, cholesterol, and DSPE-PEG2000. This was explained by the lower uptake of PSL by the liver and spleen compared to nPSL [143]. Similarly, PSL-DOX resulted in higher nuclear accumulation of DOX in HeLa cells in vitro. This nuclear accumulation was associated with higher activation of cleaved caspase-3 [144].

The specific accumulation of PSL in the tumor is based on the EPR effect due to the PEGylation that prolongs the circulation time. However, the selectivity of PSL can be improved through tumor-specific ligands that bind to overexpressed cell surface receptors on the tumor cells. Silva et al. coated the surface of PSL with folic acid to exploit the upregulation of folate receptors in cancer cells. The biodistribution study revealed higher DOX accumulation in the tumor 4 h after the administration of folate-coated PSL-DOX into 4T1-bearing mice compared to non-folate-coated PSL-DOX. Consistent with this finding, the highest tumor growth inhibition was observed in mice treated with folate-coated PSL-DOX. Furthermore, treatment with folate-coated PSL-DOX resulted in fewer metastatic foci in the lungs in comparison with free DOX and non-folate-coated PSL-DOX. Despite the morphological changes detected by the histological analysis of the heart (i.e., cardiomyocyte vacuolization), no prolongation in the QT interval was observed in mice treated with folate-coated PSL-DOX [142]. In another study, Sonju et al. conjugated DOX with a peptidomimetic ligand that has selectivity toward HER2-overexpressed cancer cells. The peptidomimetic-DOX (5-DOX) conjugate was delivered to the tumor by a PSL composed of DOPE, CHEMS, cholesterol, and DSPE-PEG in a molar ratio (5.8:3.7:4:0.25). This formulation released ≈60% of 5-Dox within 5 h of incubation at 37 °C in PBS buffer (pH = 6.5). In vitro, the prepared PSL (PS5-DoxL) was significantly more effective in tumor growth inhibition in HER2-positive lung and breast cancer cell spheroids than free DOX. Similarly, Calu-3-bearing mice had the smallest tumors after treatment with PS5-DoxL. Importantly, The least morphological changes in the heart and spleen were observed in PS5-DoxL-treated mice [145].

Recently, a novel PSL (pH-lipo-Dox-Pba) has been developed, combining both DOX and pheophorbide-a (Pba), a photosensitizer that can target different organelles in the cytosol upon laser irradiation. The main aim of this strategy was to target both the nucleus and the cytoplasm of the tumor cells. Moreover, the cRGD peptide was attached to the surface of the liposome to enhance the selectivity of tumor cells. The prepared liposome released ≈60% of DOX after 24 h of incubation at 37 °C in PBS (pH = 5.5). In addition, the confocal microscopy confirmed the localization of DOX and Pba in the nucleus and the cytoplasm of 4T1 cells, respectively. However, pH-lipo-Dox-Pba irradiated with laser failed to achieve significant tumor growth suppression in 4T1-bearing mice compared to the non-irradiated liposome. This was explained by insufficient laser intensity [146].

The main advantage of PSL over TSL is the acidic TME responsiveness. This feature can simplify the application of these systems since no external stimulus is needed to activate the liposome. An external stimulus, such as HT, can complicate the clinical translation of TSL since it requires further optimization of the heating protocol (i.e., HT duration and thermal dose) [147]. However, PSLs still face considerable challenges, such as the stability and magnitude of DOX release. In particular, most PSL formulations exhibited 20–50% DOX leakage after 24 h of incubation in PBS under physiological conditions (i.e., 37 °C, pH: 7.4) [142,143,145]. Furthermore, the complete reliance of PSLs on the EPR effect is still a drawback of this approach due to the reasons discussed above (Section 2). 

## 6. Conclusions

Despite the advances in nanoparticle drug delivery, the optimal balance between efficacy and toxicity of DOX has not been achieved in the clinic yet. The approved liposomal formulations (i.e., Doxil and Myocet) improved the safety profile of DOX without superior efficacy. The functionalization of the PLD surface with targeting ligands enhanced the efficacy of the liposome in the preclinical models. Among the modified PLDs, two antibody-conjugated PLDs reached clinical trials. However, the large size of the antibodies can limit their penetration. Therefore, other molecules such as aptamers, carbohydrates, peptides, and cell-penetration enhancers (CPEs) have been investigated for PLD modification. Recently, CPEs have gained great attention in this field due to the possibility of using them alone or with other targeting ligands for dual functionalization. The reviewed studies demonstrated that functionalized PLD could enhance the anti-tumor activity in rodents compared to PLD. However, challenges such as the optimal ligand density and the fast clearance of some functionalized PLDs need to be addressed. 

Thermo-sensitive liposomes (TSLs) offer the most promising approach for more efficient DOX delivery to the tumor at mild hyperthermic temperatures (39–42 °C). The main drawback with the traditional TSLs is the unsatisfactory rate and magnitude of the release of DOX, which can be enhanced by the inclusion of small amounts of lysolipids into the liposome membrane (e.g., LTLD). Although LTLD failed in two phase III clinical trials, it is the closest TSL formulation for FDA approval. However, modifications in the design of those trials are required. An alternative approach to LTLD is the polymer-modified TSL (pTSL) which can eliminate the undesired leakage observed with LTLD at physiological temperature. However, the optimization of their temperature responsiveness and hyperthermia duration is needed. Phosphatidylglycerol-based TSL (DPPG-TSL) offers an optimal approach to deliver DOX due to its stability and rapid release profile. Recently, DPPG-TSL has become closer to clinical translation after promising results in large animal models. Finally, surface functionalization and thermosensitivity can be combined in the multifunctional TSL for more efficient delivery of DOX. However, multifunctionality can increase the complexity of this approach and affect its stability and safety in vivo. Apart from TSL, pH-sensitive liposome (PSL) offers another promising approach to deliver DOX to the tumor without applying external stimulus. However, the optimization of the stability and DOX release rate is needed.

## Figures and Tables

**Figure 1 pharmaceutics-15-00893-f001:**
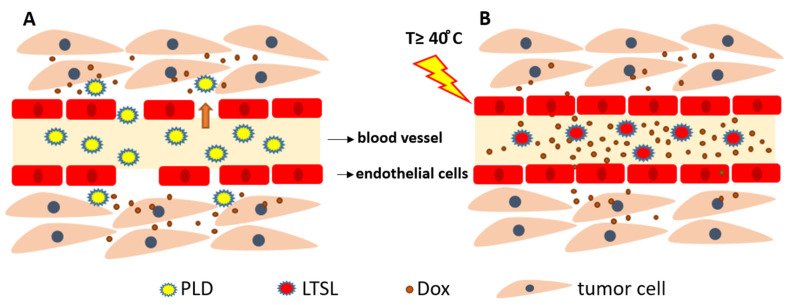
Interstitial and intravascular doxorubicin (DOX) release. (**A**) PEGylated liposomal doxorubicin (PLD) extravasates through the gaps in the tumor vasculature and release DOX in the tumor interstitium. (**B**) Lyso-thermosensitive liposome (LTSL) releases DOX in the blood vessels in the tumor followed by DOX diffusion into tumor interstitium (based on Kneidl et al. [34]).

**Figure 2 pharmaceutics-15-00893-f002:**
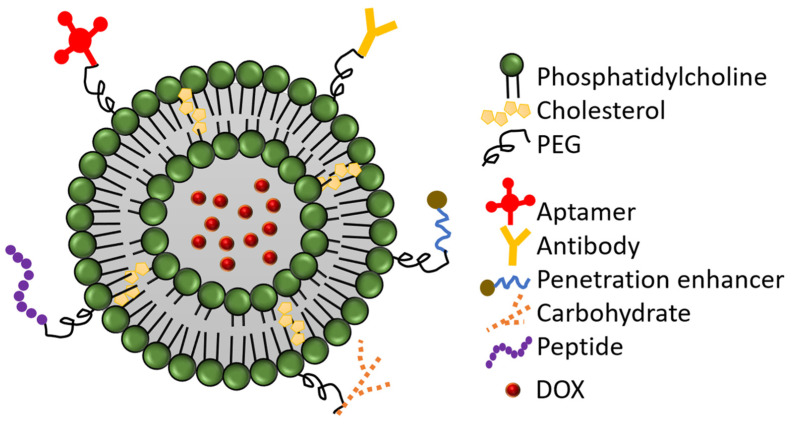
Modifications of PEGylated liposomal doxorubicin (PLD) surface by ligation of different molecules to the terminal end of PEG. (based on Khan AA et al. [38]).

**Figure 3 pharmaceutics-15-00893-f003:**
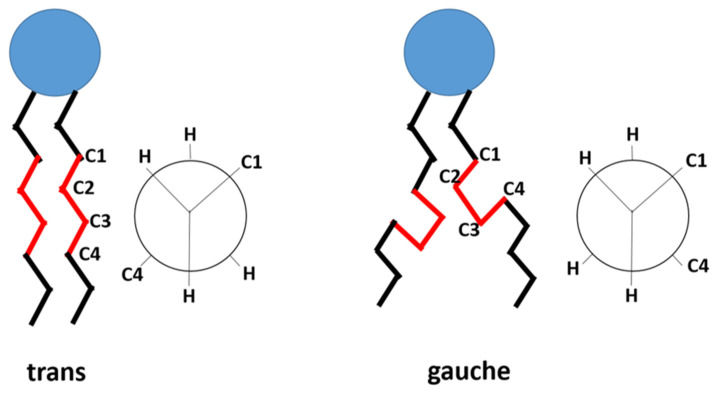
The trans and gauche conformers in the hydrocarbon chain of the phospholipid. In gel phase, trans conformers are predominant. In liquid-crystalline phase, both gauche and trans are present. (based on Kuc et al. [82]).

**Figure 4 pharmaceutics-15-00893-f004:**
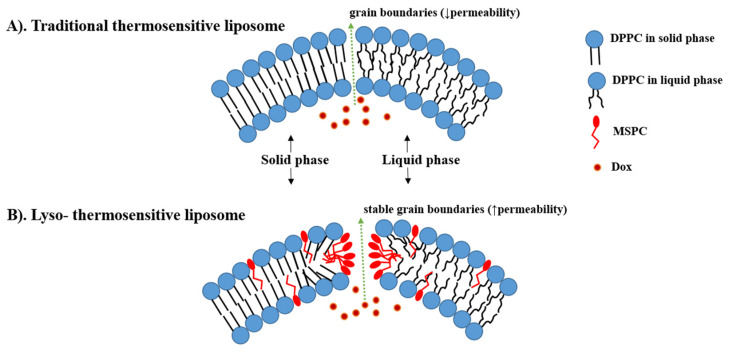
The difference between traditional thermosensitive liposomes (TTSL) (**A**) and lyso-thermosensitive liposome (LTSL) (**B**) during the phase transition. The lysolipids stabilize the grain boundaries resulting in rapid DOX release. (based on Ta et al. [83]).

**Figure 5 pharmaceutics-15-00893-f005:**
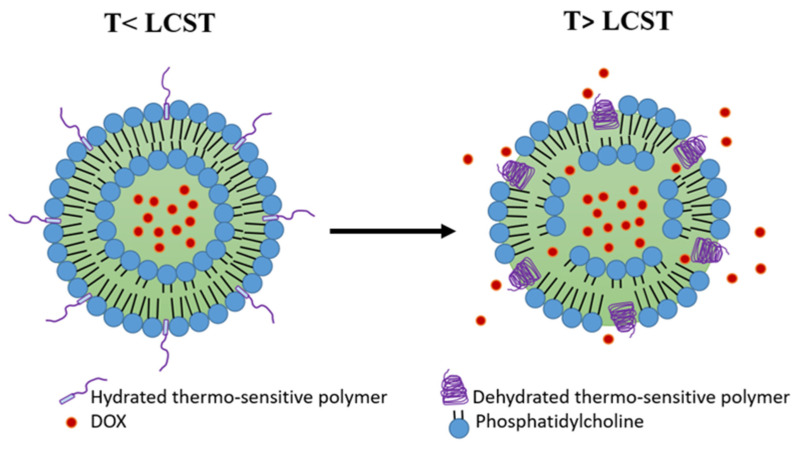
Polymer-modified thermosensitive liposome (pTSL). At temperatures below the lower critical solution temperature (LCST), the polymers are in a hydrated coil form and the membrane is stable. At temperatures higher than LCST, polymers become dehydrated and globular, destabilizing the TSL membrane and releasing DOX. (based on Mo et al. [120]).

**Table 1 pharmaceutics-15-00893-t001:** Summary of the preclinical and clinical results of the recently functionalised-PLD.

Conjugated PLD	Conjugate Type	Preclinical Results (Compared to PLD)	Year	Clinical Results/Trial Status/Possibility to Advance to Clinical Trial
MM-302	antibody	Improved anti-tumor activity in mice [45] Effective on metastatic lesions [33]	1997 [77]	Phase I: safe in a dose 8–50 mg/m^2^ + less PPE than PLD [43] Phase II (NCT02213744): terminated in 2017 No new trials registered on clinicalTrials.gov
C225-ILS-DOX	antibody	Improved anti-tumor activity in highly multidrug-resistant mouse model [50]	2003 [78]	Phase I: safe in a dose 50 mg/m^2^ + no cardiotoxicity or PPE [44]
				Phase II (NCT02833766) completed in 2020, no results published
				Phase I (NCT03603379): no DOX in CSF + high levels of DOX in glioblastoma tissues No new trials registered on clinicalTrials.gov
anti-CD147 ILs-DOX	antibody	Affected the stemness of HCC cells in vitro	2018	Further investigation for the accumulation in the lung 72 h after treatment
		Improved DOX accumulation in the tumor and the anti-tumor activity in mice [52]		
LP16-PLD	peptide	Improved anti-tumor activity in mice [57]	2018	Further optimization for ligands density is needed
LP31-PLD	peptide	Improved anti-tumor activity and survival rate in mice [56]	2021	Further optimization for ligands density due to the accumulation in the lung (LP-PLD 200)
AHNP-PLD	peptide	Improved anti-tumor activity and survival rate in mice [58]	2016	Further optimization for ligands density is needed
SS-02-PLD	peptide	Higher caspase activity in vitro Improved anti-tumor activity and survival rate in mice No improvement in DOX accumulation in tumor [59]	2021	Further studies to prove the enhanced accumulation of DOX in tumor
LinTT1-PLD	peptide	Higher cytotoxicity in 3D breast cancer spheroids [60]	2021	In vivo validation of the results is needed
PLD-P700	peptide	Higher cellular uptake and cytotoxicity in vitro [36]	2020	In vivo validation of the results is needed
cRGD-PLD	peptide	Enhanced cytotoxicity in vitro [61]	2022	In vivo validation of the results is needed
5TR1-PLD	DNA aptamer	Enhanced anti-tumor activity and survival rate in mice [63]	2018	possible. Further studies in non-rodent species (dogs or pigs)
Syl3c-PLD	DNA aptamer	Enhanced DOX accumulation in tumor and survival rate in mice [64]	2020	Further in vivo studies are needed as the current study did not show significant difference on tumor growth inhibition between Syl3c-PLD and PLD
TSA14-PLD	RNA aptamer	Enhanced tumoral DOX and tumor growth inhibition in mice [35]	2016	Further studies in non-rodent species (dogs or pigs)
ER-lip	RNA aptamer	Enhanced anti-tumor activity, DOX accumulation in tumor and survival rate in mice [65]	2021	Further studies in non-rodent species (dogs or pigs)
DOX-MFPL	Carbohydrate	Enhanced anti-tumor activity and DOX accumulation in tumor in mice [67]	2019	Further studies to evaluate DOX biodistribution to liver, spleen, lung and kidney
TAT-PLD	CPE (linked to PEG2000)	Enhanced cellular uptake in vitro Low anti-tumor activity and survival rate in mice [71]	2016	Not possible. TAT-PLD did not show any improvement in tumoral DOX, anti-tumor activity and survival rate
TAT-PLD	CPE (linked to PEG1000)	Improved anti-tumor activity, DOX accumulation in tumor and survival rate in mice [72]	2021	Further studies to optimize TAT density on the surface of PLD as different densities did not show the same results in 2 mouse models
DAG-Arg4-PLD	CPE	Improved anti-tumor activity in mice Did not improve DOX accumulation in tumor [70]	2021	Further studies to evaluate DOX biodistribution as DOX accumulation was low in the tumor but high in the lung.
Dual DOX-L	CPE + transferrin	Improved DOX delivery to the nucleus in vitro Improved DOX accumulation in tumor and anti-tumor activity in mice [73]	2018	Further studies in non-rodent species (dogs or pigs)
D31-PLD	peptide + CPE	Improved anti-tumor activity in mice Low DOX accumulation in tumor due to fast clearance [74]	2022	Not suitable for DOX delivery due to the fast clearance and high accumulation in liver and spleen
PLD-100	CPE	Improved anti-angiogenic properties in vitro Improved anti-tumor activity and median survival time in mice [75]	2020	Further studies to evaluate the anti-angiogenic effect in vivo

**Table 2 pharmaceutics-15-00893-t002:** Summary of the clinical trials on lyso-thermosensitive liposomal doxorubicin (LTLD).

Trial ID/Name	Status	Phase	Disease	Intervention
NCT03749850/i-GO	recruiting	I	breast cancer	LTLD + Cyclophosphamide + MR-HIFU
NCT02536183	recruiting	I	relapsed/refractory solid tumors	LTLD + MR-HIFU
NCT02112656/OPTIMA	completed	III	HCC	LTLD + RFA vs. RFA ( ≥ 45 min)
NCT00617981/HEAT	completed	III	HCC	LTLD + RFA vs. RFA (12–60 min based on tumor size)
NCT02181075/TARDOX	completed	I	Liver tumors	LTLD + Focused Ultrasound
NCT00826085/DIGNITY	completed	I/II	breast cancer	LTLD + Microwave HT (60 min)
NCT00441376	completed	I	HCC	LTLD + RFA

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
