# Peer review of "Recent Preclinical and Clinical Progress in Liposomal Doxorubicin"

_pharmaceutics, 2023, doi:10.3390/pharmaceutics15030893_

Round 1

Reviewer 1 Report

This manuscript concludes several strategies that can enhance the efficacy and safety profile of liposomal doxorubicin which used to tumor, and reviews the recent preclinical and clinical progress. Those latest formulations such as TSL and modified PLD represent promising DOX delivery systems with great potential for clinical impact in the near future. However, some points of the manuscript should be improved. Specific comments are given below.

1.    At lines 185, it is recommended to give some instances to briefly introduce limitations.

2.    At lines 284, it is recommended to add information about the absence of cardiotoxicity.

3.    At lines 287, the definition is more consistent with cell-penetrating peptides (CPPs), it is suggested to supplement the definition of cell penetration enhancers (CPEs).

4.    Some units are not properly marked, such as mg/m2 at lines 134, 137 and 154, -NH2 at lines 254, per m2 at lines 627 and Fe3O4 at lines 659.

5.    Please notice the title sorting, lines 270 should be 3.4 and lines 286 should be 3.5.  And the title format should be unified, the liposome in 4.2 and 4.3 are singular (liposome), while in 4.1, 4.4 and 4.5 are plural (liposomes).

6.    At lines 211, there has an extra e in HER2e/Neu which should be removed.  At lines 112, the colon should be removed. At lines 390, there should add a closing parenthesis. Please carefully check the manuscript for spelling and grammar.

Reviewer 2 Report

Review «Recent preclinical and clinical progress in liposomal doxorubicin: surface functionalization and thermosensitivity»

For last decades liposomes are intensively used as biocompatible and biodegradable containers for encapsulation of antitumor drugs. This paper describes the recent trends in synthesis of liposomal doxorubicin (DOX), since DOX has garnered great interest as anti-cancer due to its high efficacy despite dose-limiting toxicities. 

This Review is of substantial interest, sufficiently original, and the presented information is important in the field of nanopharmaceutics. The abstract reflects briefly the principal idea of the paper. Introduction provides a good manuscript`s background.

I recommend Review for publication after addressing the following:

1.    In section 3.2. the authors discuss the Carbohydrates- conjugated pegylated liposomal doxorubicin.  In my opinion, it would be helpful to mention investigations on conjugates of liposomes with chitosan in this part of the review.

2.    It would be also helpful to review the potential DOX-loaded pH-sensitive liposomal formulations and evaluate their advantages and disadvantages compared to thermosensitive ones.

3.    The authors should enter different numbers for subsections named “Peptide-conjugated PLD”, “CPE- conjugated PLD” and “Carbohydrates- conjugated PLD”. In the text of the manuscript these sections have the identical number 3.2.
